# Effects of Cigarette Smoking on Influenza Virus/Host Interplay

**DOI:** 10.3390/pathogens10121636

**Published:** 2021-12-17

**Authors:** Jerald Chavez, Rong Hai

**Affiliations:** Department of Microbiology and Plant Pathology, University of California-Riverside, Riverside, CA 92521, USA; jchav033@ucr.edu

**Keywords:** influenza virus, smoking, host response

## Abstract

Cigarette smoking has been shown to increase the risk of respiratory infection, resulting in the exacerbation of infectious disease outcomes. Influenza viruses are a major respiratory viral pathogen, which are responsible for yearly epidemics that result in between 20,000 and 50,000 deaths in the US alone. However, there are limited general summaries on the impact of cigarette smoking on influenza pathogenic outcomes. Here, we will provide a systematic summarization of the current understanding of the interplay of smoking and influenza viral infection with a focus on examining how cigarette smoking affects innate and adaptive immune responses, inflammation levels, tissues that contribute to systemic chronic inflammation, and how this affects influenza A virus (IAV) disease outcomes. This summarization will: (1) help to clarify the conflict in the reports on viral pathogenicity; (2) fill knowledge gaps regarding critical anti-viral defenses such as antibody responses to IAV; and (3) provide an updated understanding of the underlying mechanism behind how cigarette smoking influences IAV pathogenicity.

## 1. Introduction 

As of 2018, the CDC estimates that current cigarette smokers represent 14% of the US population, representing 34 million Americans [1]. Cigarette smoking results in 480,000 annual deaths in the United States and is estimated to have resulted in over ten times the number of premature deaths of all US fought wars combined [1]. Smoking is a well-known cause of pulmonary conditions such as chronic obstructive pulmonary disease (COPD), directly affecting risk and degrees of symptoms. However, besides the direct damage cigarette smoke (CS) can inflict on the pulmonary system, it is also a well-known risk factor for the development and exacerbation of infectious diseases such as influenza virus [2,3,4,5]. The CS specific mechanism(s) directly responsible for the impact on risk and disease outcome remains unclear. A direct complicating factor for this study of CS induced disease and complications is that CS is comprised of over 4000 different compounds and toxins. This high number of chemicals makes isolation of a single causative agent for pathologies induced by CS extremely difficult and time-consuming. Additionally, human genetic variability and the fact that smoking can result in other chronic diseases adds further confounding factors to the study of CS exposure and infection in human populations. The etiology of smoking’s impact on infection outcomes remains relatively understudied. Despite this hurdle, a common thread in the CS induced exacerbation of chronic and infectious disease appears to be acute and chronic (dubbed “low-grade”) inflammation. Chronic inflammation is a well-studied risk factor in the development of chronic diseases such as cardiovascular disease (CVD). Critical immune responses necessary to combat infection depend on induced inflammation, yet it is unclear how or if CS-induced acute and chronic inflammation directly affects infectious disease, or where these sources of inflammation would be coming from. In this review, we will examine the known impact of CS exposure on immune responses critical to influenza infection, the exacerbation of inflammation (local and systemic) during infection in smokers and smoking models, potential CS induced changes in virus replication due to changes in immune response, and potential impacts of low-grade inflammation on influenza disease outcomes.

## 2. Influenza Virus and Response to Normal Infection

A review of the general life cycle of influenza viruses and how the immune system normally responds to infections is helpful to understand how influenza infection is affected by cigarette smoking. Influenza viruses are negative sense, segmented, RNA, enveloped viruses and part of the Orthomyxoviridae family. There are four types of influenza virus: Type A, B, C, and D viruses. Influenza A, B, and C all infect humans, with type D primarily infecting cattle [6,7]. In humans, these viruses are normally transmitted by aerosol particles, causing infection of the upper respiratory tract (nasal cavity) and in the case of severe disease, infection of the lower respiratory tract (lungs). Mild infections typically result in fever, cough, sore throat, runny nose, body aches, and fatigue. Influenza A Virus (IAV) and Influenza B Virus (IBV) are responsible for seasonal epidemics and can cause severe respiratory illness including primary viral pneumonia [8,9,10] and secondary bacterial pneumonia [11,12], especially in the elderly and immune compromised [13]. Together, these viruses are globally responsible for between 290,000 and 650,000 deaths annually [14] with the economic burden in the US estimated to exceed $14 billion yearly as of 2018 (healthcare + lost productivity estimates combined) [15]. 

In response to infection, the body initiates both an innate and adaptive immune response, in which the non-specific innate response restricts and contains infection, while the targeted adaptive response is responsible for the ultimate clearance of the virus (summarized in Figure 1). Influenza viruses first infect upper respiratory airway epithelial cells, where infection is initially detected by recognition of pathogen associated molecular patterns (PAMPs) that are inherent to the virion or produced during infection. These PAMPs, such as viral RNA, are detected by Pattern Recognition Receptors (PRRs). For example, RIG-I is an intracellular receptor (recognizes and binds the influenza viral genomic RNAs [16]), which triggers signaling cascades leading to production of the proinflammatory cytokine interferon (IFN) [17,18]. Type 1 IFNs secreted by infected cells alert neighboring cells to the infection and stimulate transcription of Interferon stimulated genes (ISGs). ISGs in turn produce an antiviral state that restricts virus replication. The release of pro-inflammatory cytokines and chemokines results in the recruitment of circulating immune cells from the blood to the site of infection. NLRP3, activated by multiple Influenza PAMPs [19], results in the formation of the NLRP3 inflammasome and the production of pro-inflammatory cytokines that recruit leukocytes (macrophages, lymphocytes, granulocytes, etc.) to the lungs during infection [20]. Leukocytes, such as neutrophils and macrophages, are critical to the restriction of the virus, as they serve to phagocytose infectious virus, breaking them down by lysosomal degradation. Additionally, these cells secrete additional cytokines and chemokines to sustain and increase the recruitment of other immune cells to aid in virus restriction and clearance.

Adaptive immune responses are critical for the ultimate clearance of infection. Adaptive immunity requires that viral antigens (components of the virus) be presented to adaptive immune cells for activation. The process is completed by antigen presenting cells, including dendritic cells and macrophages. They process viral proteins and present them to CD4+ helper T-cells, resulting in their activation and proliferation. These activated helper T-cells go on to activate B-cells and other effector cells, such as CD8+ T-cells. B-cells produce antibodies that bind influenza virus and prevent the virus entry into susceptible cells (neutralization), while active CD8+ T-cells seek out and kill virus infected cells. Therefore, innate immune response cells like macrophages and dendritic cells represent critical bridges from the innate to the adaptive responses.

While the recruitment and activation of innate and adaptive cells are necessary to combat infection, excess inflammation can result in severe damage to the airway and lung tissue. In severe influenza infections, viruses migrate from the upper respiratory tract to the lower respiratory tract, where infection of the lower lung epithelia triggers inflammatory cell recruitment, causing damage to the alveolar epithelial cells responsible for gas exchange in the lungs. Once infection reaches the endothelium in the interstitium, cytokine responses and inflammation are further exacerbated [21]. In some severe cases, this exaggerated inflammation and damage can lead to acute respiratory distress syndrome (ARDS) and, subsequently, respiratory failure. However, this severe disease is generally associated with pandemic influenza (such as 2009 H1N1 [22,23]) and strains of Avian influenza (such as H5N1) [24,25,26]. The exacerbated levels of pro-inflammatory cytokines and chemokines are referred to as a “cytokine storm.” This phenomenon is typically not associated with mild, but rather severe cases of influenza. Patients hospitalized with severe influenza due to infection with avian H5N1 exhibit exacerbated levels of circulating pro-inflammatory cytokines including TNF-a, IL-6, and sIL-2r, IP-10, and MIG [27]. 

Influenza virus infection eventually stimulates molecular and cellular pathways to effect tissue repair of infection damaged airways. For example, Influenza virus infection induces IL-33 expression [28]. IL-33 acts on a number of cells including but not limited to ILC2s, T_H_2 cells and T-reg cells [29], which drive the type II immune response critical for tissue repair. Thus, the immune response must stimulate enough immune cell recruitment to the local effected area to restrict or clear infection, but not so much that it results in excessive damage that slows or inhibits type II repair responses. Here, we will examine how cigarette smoking affects innate and adaptive immune responses, inflammation levels, tissues that contribute to systemic chronic inflammation, and how this affects IAV disease outcomes.

## 3. How Does CS Affect Immune Responses to IAV Pulmonary Insult?

Cigarette smoke contains both water soluble and non-soluble components. Once drawn into the lungs, they bombard the respiratory epithelial cells, triggering the release of pro-inflammatory cytokines (such as IL-8 and TNF-α) by resident immune cells and epithelial cells alike. This results in the elevated recruitment of neutrophils in acute cigarette smoke exposure and increased macrophage levels in the lungs [30,31,32,33]. Upon arrival, innate immune cells aid in the clearance of smoke particles, but also perpetuate inflammation and cellular recruitment by releasing pro-inflammatory cytokines and chemokines in the affected tissue. One could hypothesize that if inflammation and innate immunity are already elevated in smokers prior to infection, that this would aid in preventing infections, or make the clearing of infections faster for the host. However, in cigarette mouse models, IAV infection results in worse disease outcomes compared to non-smoking controls. For example, CS exposure and subsequently infection typically will result in reduced weight gain post infection [34,35,36,37,38], increased lung remodeling (deposition of scar tissue replacing functional lung tissue) [39,40] and increased mortality [33,35,36,37,41] compared to non-CS infected mice. Similarly, CS exposure in humans is shown to be associated with increased risk of influenza infections [4,42,43], increased risk of severe symptoms [4], and decreased efficacy of influenza vaccine [44,45]. 

Therefore, worse infection outcomes could be due to at least two factors: (a) CS compromises immune responses necessary for antiviral defense and thus the host incurs more direct viral damage and/or (b) IAV infection/replication is not affected, but rather a response to viral infection triggers an exaggerated inflammatory response compared to healthy individuals, resulting in more damage during infection and prolonged recovery. If the former is true, we would look for evidence that shows (a) the compromising of known immune responses necessary for containment and clearance of influenza virus; (b) evidence of more viral replication in smokers or smoking models compared to healthy individuals or controls; and (c) longer viral clearance times for CS exposed individuals vs non-CS controls. Reports do indicate that specific immune responses important to viral insult response may be altered. Hans et al. have shown that cytokines and chemokines like TNF-α, IL-6, IL-4, IL-5, IL-10, and IFN-γ levels were elevated (between 2 and 10-fold) in the mice lungs after 3 weeks of CS exposure followed by IAV infection [36]. These cytokines represent a mixture of both pro- and anti-inflammatory cytokines. For example, IL-4, 5, and 10 are classic T-helper type 2 (T_H_2) specific cytokines that normally push specific cells to commit to the T_H_2 response. 

The T_H_2 immune response function varies depending on the cells involved. Generally, they handle three broad processes: (a) response to and clearance of allergens; (b) intracellular parasites defense; and (c) tissue homeostasis and repair. Because long term smoking repeatedly introduces noxious elements to the lungs (as smoke particles) that need to be removed and cause excessive pulmonary inflammation, cigarette smoking is likely engaging both the T_H_2 allergen clearance and tissue repair mechanisms. Indeed, smoking is associated with a higher risk for allergens and asthma [46], representing engagement of the allergic T_H_2 responses. To activate the allergen response, dendritic cells (professional antigen presenting cells) uptake the allergen or noxious element and subsequently activate naive T-cells. These activated T-cells (now T_H_2 cells) migrate to lymph nodes where they further differentiate and secrete IL-4, inducing class switching in B-Cells to produce IgE specific antibodies. Additionally, these T_H_2 cells can egress from the lymph nodes and infiltrate tissue, where they can begin to produce IL-5 and IL-13. IL-5 stimulates the activation and recruitment of eosinophils, which subsequently aid in the further T_H_2 recruitment and proliferation of T-cell cytokine production (reviewed in [47]). IL-13 is anti-inflammatory, inhibiting pro-inflammatory cytokine production [48,49,50] and may represent the engagement of repair mechanisms. In smoking models of COPD, however, IL-13 is primarily responsible for the induction of emphysema [51]. 

Chronically engaging these aspects of T_H_2 immunity through smoking may likely result in a poor response to viral infection upon challenge, as the restriction and clearance of viruses like Influenza A and B viruses rely primarily on T_H_1 pro-inflammatory responses, which are antagonistic to T_H_2 responses [52]. Indeed, exogenous IL-4 administered to mice results in slower viral clearance of IAV and reduced activation of CD8+ T-cells [53], which are necessary for clearance of virus infected cells. In addition to lower activation antiviral cytotoxic T-cells, it’s possible a significant portion of B-cells in smokers are producing IgE due to engagement of T_H_2 responses. T_H_1 mediated antiviral defenses activate B-cells to produce virus specific IgG antibodies, which block virus entry into the cell (neutralization) and thus halt infections while CD8+ T-cells seek out and kill virus-infected cells to clear infection. Thus, in smokers, if T_H_2 responses were primarily engaged before infection, there may be a lag time for B-cells to undergo class switching to IgG to combat influenza virus infection compared to healthy individuals. While adaptive immune responses to IAV after CS exposure are less well characterized compared to innate responses, chronic CS exposure (>2 weeks) leads to decreased IFN production by CD4^+^ and CD8^+^ T cells [35], no change to serum (IgG) IAV-specific antibodies 6 weeks post infection [41], but decreased levels of IAV IgA [37,54]. Effector T-cells secrete cytokines like IFN- in order to carry out their function. Lower IFN production could be predictive of decreased viral clearance and lower activation of subsequent adaptive immune cells like B-cells. No change in serum antibodies in CS exposed models compared to non-CS groups could suggest that if skewing of adaptive immunity is occurring, it is at a level that does not have a significant impact on IgG IAV antibodies. However, it should be noted that the kinetics of antibody responses to IAV infection in CS models to our knowledge has not been done, potentially missing critical early effects of smoking on serum antibody responses. Reduced levels of IAV IgA antibodies on the other hand indicates at a minimum that smoking is having an impact on mucosal immunity. IgA antibodies are secreted into the mucosal lining of the respiratory tract, where they can intercept their viral target prior to it reaching the epithelium, preventing infection. As such, a decrease in IgA response in CS models during Influenza infection represents a direct compromise of the viral immune response [55].

So, if immune responses necessary for the response to and clearance of Influenza are compromised, the follow-up question would be whether viral replication has been affected? WSN IAV had higher infectivity levels in human small airway epithelial cells exposed to cigarette smoke compared to air exposed controls [55]. Similarly, Calu-3 cells (lung epithelial cell line) exposed to CS supported 1-log higher replication of H1N1 IAV compared to non-CS controls [56]. Comparing viral replication in nasal epithelia cells isolated from smokers versus non-smokers, it was reported that there was a six-fold higher virus replication in cells from smokers compared to healthy controls [57]. However, mouse model data regarding viral loads with CS exposure are less compelling. Gualano et al. reports 4-day CS exposure increases in H3N2 viral loads compared to controls, though only to a moderate degree with a ~3.5 fold increase [34]. Others groups report CS exposure having little to no effect on viral loads [33,35,41,58]. This may suggest that a worse disease outcome may result from other factors (such as exaggerated inflammation) rather than compromised immune responses in smokers. However, it should be noted that these results may likely reflect a lack of standardization in model systems, as CS exposure length, cigarette type and number, mouse genotype background, IAV infection MOI, and IAV strain used do not have any standard in the field. For example, CS exposure in mouse models of infection can range from 3 days [59] to 6 months [54] prior to infection. 

If viral loads are not significantly affected by CS exposure, then could clearance of IAV infection be compromised? Mebratu et al. showed, using mouse adapted H3N2- HKx31 at 1 × 10^3^ PFU/mouse with up to 1 month CS exposure, that CS exposed mice had 1 log higher titers at 14 dpi (as measured by qPCR) compared to air control mice [60]. Similarly, Hong et al. noted that, with H3N2 at 25 TCID50 with 3 month CS exposure, CS exposed mice had 100 fold higher live virus titers at 10 dpi vs. the no CS controls, suggesting viral clearance had been impeded [37]. Interestingly, Lee et al. has shown, using PR8 IAV at 100 PFU per mouse with 2 weeks CS exposure, that early viral titers at three days post infection were higher in CS mice vs controls, but peak viral titers and post peak clearance for CS and controls were identical in contrast to previous studies [38]. While seemingly conflicting, we would assume that these conflicting clearance results are more potentially due to variability in IAV dose, CS exposure, and viral genotype rather than being in direct contradiction. As such, it is important that further studies conducted on the effects of CS on viral loads and clearance standardize a set CS exposure amount and virus type, while titrating viral doses. This would not only determine whether CS affects viral burden and clearance with a single IAV strain, but may also help parse the effects of smoking in both mild and severe infections. 

If we examine the latter option of how CS affects infection, it is possible that, similar to chronic disease, CS exposure may exacerbate infection outcomes by exaggerated inflammatory responses to pulmonary insults (like infection). Airway epithelial cells harvested from smokers have lower levels of Type I and II IFN in response to IAV infection compared to healthy cells [57], while exposure to CS extract (water soluble contents of cigarette smoke) results in lower type I and type II IFN, IP-10, IL-6, and RIG-I transcription and expression [61,62], suggesting CS has an inhibitory effect on innate immune responses to viral insult in humans. Oddly, these results suggest the opposite of the proposed exaggerated response mechanism. However, the length of years smoking and the packs smoked per day may have a significant impact on local and systemic inflammation responses. In mice, response to IAV infection with sub-chronic CS exposure (<2 weeks) appears to have a suppressive effect on a range of cytokines (TNF-α, IL-1B, IL-6, and IP-10) and are reported to have increased BALF and lymph node CD4^+^ and CD8^+^ T cells numbers [34]. In contrast, chronic CS exposure (>2 weeks) results in exaggerated cytokine responses (TNF-α, IFN-y, IL-6, IL-12, IL-23, IL-1, IL-5, IL-10, KC, MIP-1a, IL-17, IL-1B) [36,37,40,41,54] with infection compared to non-smoking infected controls. In addition to elevated cytokine profiles, chronic exposure is also associated with increased local pulmonary inflammation including increased lung tissue and BALF neutrophils [34,37,38,40,41,58], macrophages [34,38,40,41,60,63], and lymphocytes [40,60,63] compared to non-CS exposed animals. 

In summary, there is evidence that innate and adaptive responses to influenza infection are altered in CS patients, which was confirmed in CS animal models. However, reports are inconclusive as to whether CS exposure causes a higher than normal viral burden or longer than average viral clearance. Irrespective of the virological response, there are increased pro-inflammatory cytokine profiles in chronic CS exposure models, with increased pulmonary cellular recruitment in acute and chronic CS exposure. This reflects an exaggerated response to infection, in which IAV infection in chronic CS models results in increased pro-inflammatory cytokine production, and subsequently exaggerated recruitment of inflammatory immune cells. These will ultimately cause more tissue damage and worse disease outcomes of infection compared to non-CS individuals. Conflicting data on virological responses most likely reflect a lack of standardization for multiple experimental parameters within CS models, meaning care should be taken when interpreting conclusions about virological responses, and as such, a more thorough investigation using standardized IAV strains and CS exposure to directly compare viral burden, clearance results, and achieve a relative consensus is needed. 

## 4. Cigarette Smoke, Chronic Inflammation, and the Crossroads of Infection with Chronic Disease

Besides the acute effects CS has on immune inflammation in the lungs, CS exposure also results in systemic, sustained chronic inflammation. Acute inflammation, driven by infection or tissue damage, is short term in nature. As described above, normal immune response to IAV infection is generally resolved by cells at the site of infection. This is initiated by cells releasing pro-inflammatory cytokines and chemokines, activating and recruiting circulating immune cells to the area of infection to restrict and ultimately clear infection. Typically, 2–3 weeks after infection, the infection is cleared and tissue inflammation recedes, followed by tissue repair and pro-inflammatory markers returning to normal pre-infection levels. In contrast, chronic inflammation is non-resolving inflammation which persists for long periods of time, as the name would suggest [64]. This type of inflammation is associated with elevated levels of circulating inflammatory cytokines and acute phase proteins. For example, C-reactive protein serum levels in chronic obstructive pulmonary disease (COPD) patients are roughly twice that of non-smoking patients (~4.0 mg/L vs 1.9 ml/L) [65,66]. This elevation in circulating inflammatory factors is likely the result of oxidative stress induced by reactive oxygen species (ROS) that are inhaled from the gaseous phases of CS [67,68,69,70,71,72,73]. Oxidative stress refers to an imbalance between reactive oxygen species (ROS) and antioxidants within a cell. ROS production normally occurs under physiological conditions as part of normal oxygen metabolism. However, high concentrations of ROS can result in harmful reactions with multiple macromolecules including nucleic acids and cellular membrane lipids leading to cell and tissue damage. This cellular damage induces pathways that result in elevated pro-inflammatory cytokines like TNF- & IL-8 [74]. Smoking created a sustained inflammatory cycle by the constant inhaling of toxins like ROSs, which cause damage to tissues and induces pro-inflammatory cytokine/chemokine production, recruiting more inflammatory cells that further release more inflammatory signals, perpetuating the cycle further (Figure 2).

This sustained inflammation could have a desensitizing effect on cellular responses, potentially including those that are responsible for viral containment and clearance. For example, human monocytes (white blood cells that can differentiate into macrophages and dendritic cells) cultured with Lipopolysaccharides (LPS, bacterial endotoxin) produce less TNF-α when subsequently stimulated with LPS compared to controls even when LPS receptors’ expression has increased, suggesting a negative feedback loop that suppressed TNF-α production [75]. To better examine CS induced chronic inflammation and how it may affect infection, we first wish to draw parallels to effects of CS low-grade inflammation in chronic disease. This examination may shed light on sources of low-grade inflammation and provide some mechanistic details not currently known in the CS-influenza field. To begin with, we will examine a chronic disease, cardiovascular disease, by defining it, and exploring how CS exposure affects the inflammation potentially involved in its pathology.

## 5. Contribution of Adipose to Cardiovascular Disease and Chronic Inflammation

Cardiovascular disease (CVD), or heart disease, is an umbrella term referring to multiple heart related conditions, including coronary artery disease (CAD), arrhythmia, cardiomyopathy, and atherosclerosis. Cigarette smoking is a substantial risk factor for the development of CVD [76]. CVD patients who are current smokers are at increased risk of cardiovascular events with higher mortality compared to non-smokers [77]. CVD results from the narrowing or blocking of blood vessels that can result in severe chest pains, heart attack, and stroke (known collectively as cardiovascular events). This narrowing occurs from deposition of lipids, cell debris, connective-tissue components, and whole cells that form plaques along the lining of blood vessels. Tissue inflammation can lead to increased cell death (elevating circulating cell debris) and increased inflammatory cell recruitment to the epithelium of blood vessels, increasing the risk of blockage and plaque formation. So how does CS create the environments necessary for sustained, low-grade inflammation?

Adipose tissue may serve as an additional source of excess inflammation due to CS exposure. Adipose tissue can be divided into either a) “white” adipose tissue that is for energy storage/cold insultation or b) “brown” adipose tissue which releases energy. White adipose tissue can be subdivided into either subcutaneous adipose tissue (SAT) which resides just below the skin, or visceral adipose tissue (VAT) that surrounds the internal organs [78,79]. Lear et al. (2012) used a linear regression model to compare European and South Asian groups (200 each) to determine if body fat, SAT or VAT mediated increased risk of CVD in South Asian populations despite comparable body mass indexes (BMI’s) between the two populations. They found that VAT was the highest contributing factor to CVD risk, contributing between 16%–52% of the difference in CVD risk between groups [80]. Other studies have noted that smoking is associated with higher VAT deposits despite overall less total fat compared to obese individuals [81,82]. Subsequently, Terry et al noted that, when adjusting for obesity and other factors like physical activity and diet in a 3000 person dataset for 18–30 year-olds, smokers had lower BMI’s compared to non-smokers, but higher abdominal VAT deposits and intramuscular fat [83]. In healthy individuals, adipose tissue contains 5%–10% macrophages. However, in chronic inflammatory conditions like obesity, perirenal/mesenteric/subcutaneous adipose tissues can contain upwards of 50% macrophages in mouse models of obesity [84]. Macrophages are polarized and exist on a spectrum from M1-like pro-inflammatory macrophages to the M2-like anti-inflammatory macrophages. TNF-α along with INF-γ polarize macrophages towards the M1 pro-inflammatory state [85]. Because Terry et al utilized a historical dataset, they could not measure cytokine conditions in the VAT tissue itself to confirm pro-inflammatory cytokine profiles but given serum C-reactive protein (blood marker of systemic inflammation) levels were elevated in smokers compared to non-smokers, it likely suggests a state of systemic inflammation. Therefore, CS systemic inflammation may have ramifications for any organ that is surrounding by VAT, including the heart. 

Excess epicardial adipose tissue (EAT) is associated with increased risk of CVD [86,87]. EAT is a type of VAT surrounding the heart and is in contact with major coronary artery. This heart tissue normally functions under homeostatic conditions to both buffer the heart against the mechanical force of pumping as well as to serve as a secretory tissue supplying anti-inflammatory molecules (like IL-4, IL-10) to the surrounding tissue, suppressing inflammation, and preventing excess plaque buildup. Smoking is associated with increased EAT volumes [88,89,90], and EAT biopsied from current cigarette smokers undergoing scheduled cardiac surgery had significantly elevated levels of TNF-α and IL-6 pro-inflammatory cytokine protein levels compared to non-smokers [91]. This pro-inflammatory state may have significant consequences for resident immune cells. While Mach et al did not check for INF-γ levels [91], the presence of TNF-α in higher concentrations than normal may suggests that EAT macrophages have skewed towards the M1-like state due to CS exposure. It’s possible that this inflammatory state in heart EAT is driven at least in part by CS induced oxidative stress. Itoh et al noted that in a mouse model of acute CS exposure (10 days of CS exposure), that while they did not notice a difference in macrophage infiltration due to CS exposure, they did note that there was an elevation in oxidative stress in both the heart VAT and groin adipose tissue of the CS mice compared to non CS mice [92]. This could result in a higher risk of immune cell recruitment. 

In summary, smoking exposes airway tissue to over 4000 different chemicals and compounds, including ROS. This leads to oxidative stress and excess tissue inflammation. Another pathogenic effect is the accumulation of VAT tissue around organs. Furthermore, the excess VAT tissue is likely to undergo inflammation, which will complicate the disease outcome. 

## 6. Adipose Tissue and IAV Infection

If smoking is resulting in chronic inflammation due to excess organ adipose tissue like EAT, what are the consequences, if any, on infection? Obese models of infection may provide some insight. Adipose tissue from obese individuals chronically secretes elevated levels of pro-inflammatory cytokines (most likely because of greater immune cell infiltration) [93]. Leptin upregulates pro-inflammatory cytokine production and phagocytosis in monocytes [49,50], while in T-cells, it normally stimulates proliferation. However, in the hyperleptinemic state, T-cells can become resistant to leptin in rodent obese models [94]. At the least, this suggests that CS overstimulation and chronic production of cytokines could result in desensitization of critical IAV response cell types. Obesity is associated with excess adiposity, and adipose cytokines (“adipokines”) like TNF-α, IL-6, and MCP-1 are elevated in obese mouse models [95,96], suggesting low grade inflammation is present similar to CVD. Respiratory epithelial cells isolated from obese individuals supports higher IAV replication but expresses lower IP-10 and IL-1β in response to infection compared to healthy donor cells in-vitro [97]. Primary epithelial cells collected from obese patients in culture also exhibit a delayed cytokine response to IAV infection compared to non-obese patient cells [97]. Interestingly, while these models have chronically elevated cytokine levels in adipose tissue, they have delayed cytokine profile responses to IAV infections [98], suggesting a slower immune reaction to infection, possibly due to cellular desensitization to specific cytokines. Similar to CS models, obese mice have a delayed, but exaggerated immune inflammation profile upon infection with IAV. It is possible that CS induced adipose tissue is adding to the systemic inflammation already ectopically present in chronic diseases such as CVD, obesity, and type-2 diabetes. This would not only exacerbate these conditions by increasing the secretion of pro-inflammatory cytokines but would also add to a dampening/suppressing effect on the ability of cells to respond in a timely fashion to influenza infection. For example, cytokine receptors after interaction with their agonist, can either be recycled to the cell surface or degraded, desensitizing the cell from the signal [99]. It’s possible that chronically elevated cytokines produced from a number of sources like adipose tissue due to CS exposure induces chronic inflammation, potentially de-sensitizing non-immune and immune cells to specific signals for infection (Figure 3). Mouse and rat model systems used in CS IAV infections mentioned previously utilized CS exposure ranges from 2 weeks to 6 months or longer but were not designed to examine adipose deposits nor divorce their potential impact on infection response from local pulmonary inflammation. Therefor in CS model systems, we would ask the following: (a) when do significant VAT deposits occur during CS exposure? (b) Do these VAT deposits contribute to sustained and elevated cytokine tissue/serum levels; (c) Are tissues in contact with chronic inflammatory cytokines desensitized to molecular infection alarms (like interferon)? (d) Are immune (like macrophages) and non-immune (like lung epithelia) cell response to infection impacted by CS exposure? We speculate that CS induces chronic expression of pro-inflammatory cytokines in blood and tissue, resulting in increased VAT formation. Subsequently, CS induced VAT produces more pro-inflammatory responses, exacerbating immune and non-immune cell desensitization to standard molecular alarms to infection. As a result, these tissues would have fewer receptors for said alarms (such as IFN) and would exhibit a slower response to infection (like delayed infection cytokine responses), resulting in worse disease outcomes. 

## 7. Host Factors Involved in CS Mediated Effects on IAV Infection

Several studies have implicated host factors in the pathogenesis of CS mediated exacerbations of IAV infection outcomes. WSN IAV had greater infectivity in small airway epithelial cells when they exposed to cigarette smoke via a liquid–air interface compared to control cultures [62]. CC chemokine receptor 2 (CCR2) is a chemokine receptor that regulates monocyte chemotaxis [100]. CCR2 mouse KO ameliorates the detrimental effects of influenza infection including reduced survival and macrophage lung recruitment [101,102]. However absence of the CCR2 receptor did not attenuate the exaggerated inflammation observed in the bronchial alveolar lavage fluid (BALF), didn’t reduce weight loss or viral titers of smoke-exposed influenza-infected animals [58] suggesting that CS induces inflammation and exaggerated outcomes through occurs via other mechanisms. On the other hand, inhibition of CC chemokine receptor 5 (CCR5) resulted decreased neutrophil and macrophage levels in the airways of mice, and resulted in higher survival rates in CS exposed mice compared to non-treated groups, suggesting alleviation of excessive cell recruitment is critical for treatment of infection in smoking models [33]. CS infected mice that recovered poorly recruited γδ-T-cells expressing IL-17A. IL-17 KO mice expressed increased IFNs, made protective influenza-specific antibodies, and recovered from infection, suggesting that IL-17A activation exacerbates disease during infection [37]. MAP3K19 is a novel kinase expressed predominantly by alveolar and interstitial macrophages and bronchial epithelial cells in the lung and is overexpressed in COPD patients’ lungs. Mice exposed to cigarette smoke and influenza A virus showed a decrease in pulmonary neutrophilia, pro-inflammatory cytokines and viral load upon inhibition of MAP3K19 [59]. 

These studies represent possible pathways for immediate treatment; however, there are still significant gaps in our understanding of the combined pathology of CS and influenza virus infection. Primarily, the above studies only examine single genes or pathways. To our knowledge, previous transcriptome analysis has only focused on the how CS exposure may alter gene expression and host defense prior to infection, but not during active infection and only in specific cell types [103]. As such, we are severely limited in devising holistic strategies to combat effects of CS on infection as we do not yet understand all of the genes for which expression has been altered during infection in the presence of CS exposure (I.E. Transcriptome analysis of CS mice vs non-CS mice post IAV infection). Additionally, the genetic pathology for CS induced tissue adiposity is not clear. Development of CS induced, tissue specific, adiposity represents a source of low-grade inflammation that is not taken into account for standard anti-inflammatory treatment, much less accounted for specifically in CS IAV infections. Transcriptome level analysis would allow us to gain a more complete understanding of host defenses that have been altered or suppressed during actual infection due to CS exposure and would also shed light on the pathology of low-grade CS inflammation. An immediate first step would be to examine the transcriptome of CS mice vs non-CS mice post IAV infection in lung epithelial cells, in addition to examining the transcriptome changes that occur in organ specific adipose tissue to determine how this inflammation occurs and how it differs from local pulmonary inflammation. Finally, to our knowledge, there is no data regarding IBV pathogenesis in any smoking animal model system. Because IBV does not have an animal reservoir or known intermediate hosts, it has traditionally not been viewed to have the same pandemic potential as IAV, even though it has certain years have shown that it consistently makes similar impact to mortality and morbidity as IAV. Specifically, between the years 2000 and 2020, IBV cases represented ¼ or more of all reported cases to the CDC in the US. Therefore, an animal model is crucial for parsing how smoking can affect IBV pathogenicity and disease outcome.

## 8. Summary

CS exposure prior to influenza infection leads to higher levels of pulmonary pro-inflammatory cytokines released from resident cells, excessive recruitment of pro-inflammatory immune cells like neutrophils and macrophages, prolonged inflammation, and greater damage to pulmonary tissue compared to non-smokers and controls. There is variation in the effects of CS on IAV viral loads and clearance, ranging from no effect to moderately increasing viral loads and clearance time, depending on the study. Serum IAV IgG antibody levels post infection are not affected by CS exposure, but mucosal IgA antibody levels are lower when CS exposure occurs. As mentioned, variation between studies is most likely a reflection of significant differences in techniques in addition to the limited number of studies that examine viral loads, viral clearance, and antibody levels. As such, interpretation of how or if CS affects these aspects of viral replication or its response to it should be done with care. 

Despite technique variation, almost all studies in this field indicate that CS induces exacerbated IAV inflammation, and this is associated with worse disease outcomes and higher susceptibility to infection. Additionally, there is emerging information that shows that, while CS exposure does not appear to alter overall BMI, smoking is resulting in an increase of organ specific adiposity and chronic inflammation driven by excess specific adipose tissue. Smoking is thus leading to at least two types of inflammation, acute pulmonary inflammation, and chronic inflammation outside of the lungs, which is a known contributing factor to other chronic diseases such as CVD but has yet to be explored as a contributing factor to worse infection outcomes.

At the heart of this struggle is a severe lack of information regarding the global pathways involved in the pathogenesis of CS exacerbated inflammation during infection. For example, how is CS exposure potentially changing gene expression globally in adipose tissue? Are these changes contributing to the excess proliferation or recruitment of pro-inflammatory cells leading to chronic adipose inflammation? Similarly, CS exposure prior to infection increases the recruitment of cells such as neutrophils, macrophages, and certain T-cells to the lungs during acute inflammation to infection. Certain studies have begun to look at the activation status of specific immune cell types like T-cells during infection of smoking models, yet it is not clear if the genetic pathways necessary for their activation, proliferation, recruitment, and function have been perturbed. Studies have begun to identify host specific pathways and factors that are involved in CS induced infection pathogenesis, such as CCR2, IL-17 and MAP3K19, and thus represent exciting immediate potential therapeutic targets for treatment. Yet, in the absence of more information regarding global perturbed pathways, treatments targeting single genes or proteins may fail to achieve a reduction in CS exacerbated inflammation post infection. 

In summary, influenza virus infections in healthy individuals result in innate and adaptive immune responses that restrict and clear the virus without significant health costs to the host. In contrast, we hypothesize that cigarette smoking is increasing the risk of infection and worse disease outcomes (including higher mortality and prolonged disease) mediated by an exacerbation of innate and adaptive inflammatory responses. These responses include excessive cellular recruitment, consistent high levels of proinflammatory cytokines (similar to a cytokine storm) in the lungs and in circulation, and the potential desensitization of immune cells to specific cytokine signals necessary for mounting clearance of the virus.

## Figures and Tables

**Figure 1 pathogens-10-01636-f001:**
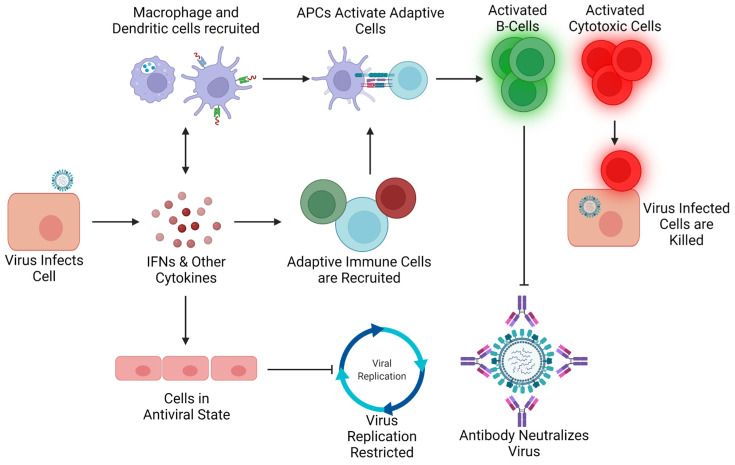
Schematic of Basic Host Defenses Against IAV Infection. Upon infection, infection is detected by intracellular receptors like RIG-I, resulting in IFN and other cytokine production. Secreted pro-inflammatory cytokines like IFNs stimulate ISG transcription resulting in an antiviral state in surrounding cells restricting virus replication. Pro-inflammatory cytokines and chemokines also recruit and activate innate and adaptive cells to the infection. Innate cells slow infection by engulfing and destroying virus particles, and present viral antigens to adaptive immune cells like helper T-cells to trigger activation and proliferation. These T-cells activate both B-cells that produce antiviral antibodies that neutralize virus and Cytotoxic T-cells that kill target and kill virally infected cells, resulting in ultimate clearance of infection.

**Figure 2 pathogens-10-01636-f002:**
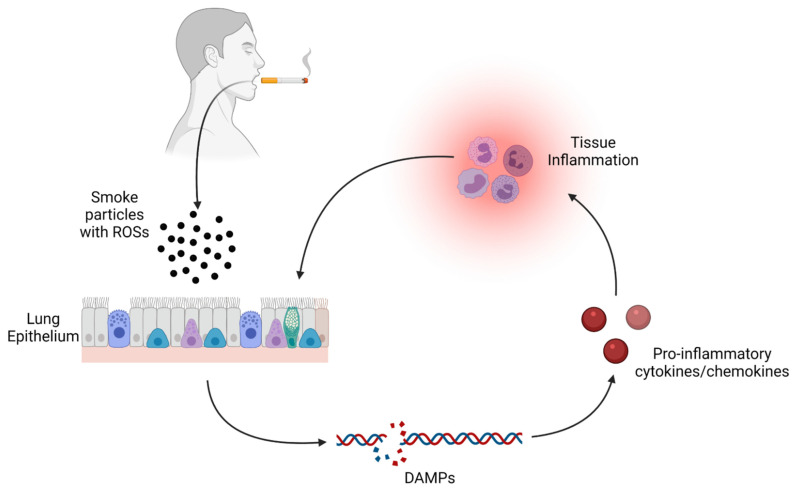
Cycle of respiratory inflammation in smokers. Cigarette smoke particles containing ROS (reactive oxygen species) are inhaled into the lungs, where cells uptake these particles causing damage to macromolecules like DNA, resulting in damage associated molecular patterns (DAMPs). These DAMPs result in pro-inflammatory cytokines and chemokines that recruit immune cells to the area, causing more inflammation and exacerbating tissue damage.

**Figure 3 pathogens-10-01636-f003:**
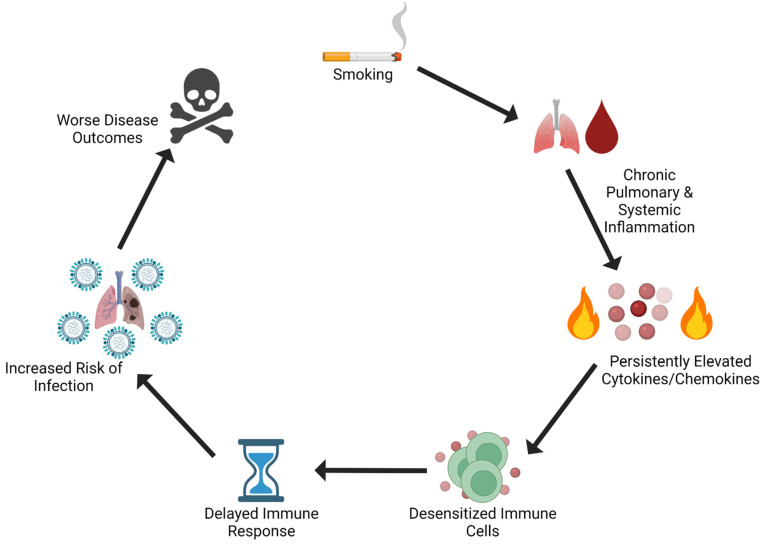
Hypothesized CS induced events and their effects on infection. Chronic CS exposure leads to persistent overproduction of pro-inflammatory cytokines, which while recruiting cells to the local area, in excess may desensitize immune cells prior to infection. Subsequently, this results in delayed response to an infection and increases the risk of infections taking hold, potentially contributing to worse disease clearance and outcomes.

## Data Availability

Not applicable.

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
