# Peer review of "Effects of Cigarette Smoking on Influenza Virus/Host Interplay"

_pathogens, 2021, doi:10.3390/pathogens10121636_

Round 1
Reviewer 1 Report
This is a well-written review focused on the effect of smoking on influenza infection. The review includes up-to-date literature and includes meta-analyses as well as experimental models. There are only a few minor suggestions to improve the quality of this manuscript:
- Figure 2, it is not clear what is ROSs. Why was that specifically included? That needs to be specified in line 300 and figure legend. Otherwise, it is confusing.
- The inclusion of figure 3 is confusing as this relates only to IAV infection and not smoke. However, it is included after (and in between) the discussion and illustration of the effect of smoke on the immune response. Overall that figure is not clear on the effect of flu in obesity either, and the legend is missing. Only the title was included.
- There are multiple typos and spaces missing between words that need to be corrected throughout the text.
Author Response
Responses to Reviewer 1’s Comments:
- Figure 2, it is not clear what is ROSs. Why was that specifically included? That needs to be specified in line 300 and figure legend. Otherwise, it is confusing.
We have modified the manuscript based on the suggestion at, Line 308.
- The inclusion of figure 3 is confusing as this relates only to IAV infection and not smoke. However, it is included after (and in between) the discussion and illustration of the effect of smoke on the immune response. Overall that figure is not clear on the effect of flu in obesity either, and the legend is missing. Only the title was included.
We have removed the figure for clarity.
- There are multiple typos and spaces missing between words that need to be corrected throughout the text.
We have corrected the manuscript accordingly.

Reviewer 2 Report
The authors have presented nice summary of effect of cigarette smoking on influenza virus infection.
- Line 32-33: change to "induced by"
- Line 78: change to "PAMPs"
- Line 107: change to "alveolar epithelial cells"
- Line 108: change "below the epithelia to "in the interstitium"
- Line 113: "Because of the inflammation damage" does this mean cytokine storm? Influneza pathology is mainly due to cytokine storm and it should be incorporated wherever the extent of damage by inflammation is being discussed.
- Line 117-122. These two sentences convey the same information. May be break up the sentences so that readers can grasp the idea easily.
- Line 127-130: The sentence is convoluted. Again break up in small sentences.
- Line 128: Change respiratory epithelia layer to "respiratory epithelial cells"
- Line 129-130: What non-immune lung cells you are referring to?
- Line 143: Change to "influenza vaccine"
- Line 146: Change to "IAV infection/replication is not affected, but rather a response to viral infection
- Line 156: Change to mice lungs instead of lung homogenate of mice
- Line 156: Change just prior to "followed by"
- Line 182: Remove infection
- Line 183: Change virally to virus
- Line 213-214: Are those cells isolated? Isolation is better word than removed.
- Line 227-228: Change to pfu/mouse
- Line 229: Change to air control mice
- Line 237: Change to "CS"
- Line 244-245: Change to inflammatory
- Line 245: Remove smoker from the start of sentence.
- Line 253: Is it defined in the literature what duration of CS exposure is considered acute, sub-chronic or chronic? Here its mentioned that subchronic is less than 2 weeks but then line number 375 mentions ten days (which are also less than two weeks) is acute CS exposure. May be this is a good point of discussion.
- Line 265: Are these inconclusive rather than conflicting? As you mentioned in line 221-225, this could be due to variation in the experimental sets up and thus clear conclusions cannot be drawn.
- Line 286: Change "with tissue repair being initiated" to "followed by tissue repair".
- Line 288-292: Does CS know to induce CRP in mice to do smokers have elevated levels? You mentioned about chronic inflammatory conditions have increased CRP but please see if there are actual numbers from smokers or mice models.
- Line 294: ROS are only produced in the endothelial cells? Or do you mean the mitochondria?
- Line 299: Change to "Smoking created a sustained inflammatory cycle"
- Line 299-300: Can ROS be inhaled directly through CS or they are produced following CS?
- Line 310: Change human monocyte cells to "human monocytes".
- Line 319: This section needs a better segway from the previous section.
- Line 333: It feels like following this sentence there will be information about how the plaques are formed. However, most of the following paragraphs talks about adipose tissue, their location and inflammatory response. As adipose tissue is not present in the vessel layers to begin with; it would be beneficial for readers to know how elevated levels of adipose tissue and inflammation leads to plaques in the lining of the vessels?
- Line 351: Ectopic adipose tissue: do you mean VAT?
- Line 377: heart: do mean oxidative stress was elevated in EAT or actual myocardiocytes?
- Line 378: Where the plaques were formed? It seems that 10 days of CS exposure is too fast to start building atherosclerotic plaques.
- Line 385: Although people have not looked at the adipose tissue levels after CS, they may have body weight charts during the exposure? Can we obtain any information on how the long term CS exposure affects body weight?
- Line 396: Change "epithelial" to "epithelial"
- Line 405: Add "that" before CS
- Line 439: Change "epithelial" to "epithelial"
- Line 441: What were those effects?
- Line 447: Neutrophil not neutrophile
- Line 474: What is local lung tissue? Lung is a tissue in itself, do you mean any other tissues nearby lungs?
- Line 509: Remove "is" and change "changing" to "changes"
Author Response
Responses to Reviewer 2’s Comments:
- Line 32-33: change to "induced by"
We have modified the manuscript accordingly, Line 34.
- Line 78: change to "PAMPs"
We have modified the manuscript accordingly, Line 79.
- Line 107: change to "alveolar epithelial cells"
We have modified the manuscript accordingly, Line 108.
- Line 108: change "below the epithelia to "in the interstitium"
We have modified the manuscript accordingly, Line 109.
- Line 113: "Because of the inflammation damage" does this mean cytokine storm? Influenza pathology is mainly due to cytokine storm and it should be incorporated wherever the extent of damage by inflammation is being discussed.
We have modified the manuscript accordingly, Line 115.
- Line 117-122. These two sentences convey the same information. May be break up the sentences so that readers can grasp the idea easily.
We have modified the manuscript accordingly, Line 123.
- Line 127-130: The sentence is convoluted. Again break up in small sentences.
We have modified the manuscript accordingly, Line 130.
- Line 128: Change respiratory epithelia layer to "respiratory epithelial cells"
We have modified the manuscript accordingly, Line 131.
- Line 129-130: What non-immune lung cells you are referring to?
We were refereeing to the epithelial cells. We have modified the manuscript accordingly, Line 132-133.
- Line 143: Change to "influenza vaccine"
We have modified the manuscript accordingly, Line 146.
- Line 146: Change to "IAV infection/replication is not affected, but rather a response to viral infection
We have modified the manuscript accordingly, Line 149.
- Line 156: Change to mice lungs instead of lung homogenate of mice
We have modified the manuscript accordingly, Line 158.
- Line 156: Change just prior to "followed by"
We have modified the manuscript accordingly, Line 159.
- Line 182: Remove infection
We have modified the manuscript accordingly, Line 185.
- Line 183: Change virally to virus
We have modified the manuscript accordingly, Line 186.
- Line 213-214: Are those cells isolated? Isolation is better word than removed.
Yes, they are isolated cells. We have modified the manuscript accordingly. Corrected, changed accordingly, Line 216.
- Line 227-228: Change to pfu/mouse
We have modified the manuscript accordingly, Line 231.
- Line 229: Change to air control mice
We have modified the manuscript accordingly, Line 232.
- Line 237: Change to "CS"
We have modified the manuscript accordingly, Line 240.
- Line 244-245: Change to inflammatory
We have modified the manuscript accordingly, Line 247-248.
- Line 245: Remove smoker from the start of sentence.
We have modified the manuscript accordingly, Line 248.
- Line 253: Is it defined in the literature what duration of CS exposure is considered acute, sub-chronic or chronic? Here its mentioned that subchronic is less than 2 weeks but then line number 375 mentions ten days (which are also less than two weeks) is acute CS exposure. May be this is a good point of discussion.
This was not a field nomenclature, but a shorthand generalized nomenclature for us based on relative effects per weeks exposed.
- Line 265: Are these inconclusive rather than conflicting? As you mentioned in line 221-225, this could be due to variation in the experimental sets up and thus clear conclusions cannot be drawn.
We agree with the suggestion and have modified the manuscript accordingly, Line 268.
- Line 286: Change "with tissue repair being initiated" to "followed by tissue repair".
We have modified the manuscript accordingly, Line 289.
- Line 288-292: Does CS know to induce CRP in mice to do smokers have elevated levels? You mentioned about chronic inflammatory conditions have increased CRP but please see if there are actual numbers from smokers or mice models.
We have modified the manuscript to include the requested information, Line 293.
- Line 294: ROS are only produced in the endothelial cells? Or do you mean the mitochondria?
We incorrectly stated specifically in the endothelial cells. We have adjusted the manuscript to clarify, Line 298.
- Line 299: Change to "Smoking created a sustained inflammatory cycle"
We have modified the manuscript accordingly, Line 303.
- Line 299-300: Can ROS be inhaled directly through CS or they are produced following CS?
The literature indeed indicates cigarette smoke contains ROS and free radicals. Besides the reference mentioned in the original manuscript, we included additional publications to strengthen this point, Line 297.
- Line 310: Change human monocyte cells to "human monocytes".
We have modified the manuscript accordingly, Line 314.
- Line 319: This section needs a better Segway from the previous section.
We have modified the manuscript accordingly. Line 322.
- Line 333: It feels like following this sentence there will be information about how the plaques are formed. However, most of the following paragraphs talks about adipose tissue, their location and inflammatory response. As adipose tissue is not present in the vessel layers to begin with; it would be beneficial for readers to know how elevated levels of adipose tissue and inflammation leads to plaques in the lining of the vessels?
Based on content of our manuscript, the plaque formation is outside the scope of this review. To clarify our point, we have adjusted the manuscript accordingly, Line 339.
- Line 351: Ectopic adipose tissue: do you mean VAT?
We meant mesenteric, perigonadal and subcutaneous adipose tissue. We have adjusted the manuscript accordingly, Line 355.
- Line 377: heart: do mean oxidative stress was elevated in EAT or actual myocardiocytes?
Clarification: The original paper was referring to VAT. We have adjusted the manuscript accordingly, Line 381.
- Line 378: Where the plaques were formed? It seems that 10 days of CS exposure is too fast to start building atherosclerotic plaques.
Plaques were not formed. We have adjusted the manuscript accordingly, Line 382.
- Line 385: Although people have not looked at the adipose tissue levels after CS, they may have body weight charts during the exposure? Can we obtain any information on how the long term CS exposure affects body weight?
Thank you for the suggestion. Based on the description of our manuscript, a “body weight charts” will be confusing to our intension.
- Line 396: Change "epithelial" to "epithelial"
We believe this means change “epithelia” to “epithelial.” We have modified the manuscript accordingly, Line 400.
- Line 405: Add "that" before CS
We have modified the manuscript accordingly, Line 408.
- Line 439: Change "epithelial" to "epithelial"
We have modified the manuscript accordingly, Line 442.
- Line 441: What were those effects?
This was an oversight. We apologize and have included the effects and associated references, Line 445.
- Line 447: Neutrophil not neutrophile
We have modified the manuscript accordingly, Line 450.
- Line 474: What is local lung tissue? Lung is a tissue in itself, do you mean any other tissues nearby lungs?
We had intended to mean the lung epithelial cells. We have modified the manuscript accordingly, Line 477.
- Line 509: Remove "is" and change "changing" to "changes"
The current sentence will be better to maintain our original intension to state that it remains elusive for the underline mechanism regarding the impact of CS on gene expression profiling in adipose tissues.

Reviewer 3 Report
Jerald Chavez et.al,
The authors systemically discussed the interplay of the chronic disease and immune response caused by influenza infection and cigarette smoking, although there is limited information about the immune response of cigarette smoking. This review will fill the immune response gaps between influenza virus infection and cigarette smoking. The manuscript is well organized and written. However, there are several areas that need to be addressed.
1. The title is not suitable for this manuscript.
2. I would like the authors to create a hypothesis model using the figure for how the immune response differences from influenza infection between CS and non-CS individuals.
3. In line 176, this is missing “,” between COPD and however.
4. In line 227, are you sure that this is the H3N3 virus?
5. In line 230, TCID50 should be “TCID50”.
6. In line 237, “cs exposure” should be “CS exposure”.
7. In line 256, the reference citation format is not consistent with other references.
8. Keep the consistent word type throughout the entire manuscript.
Author Response
Responses to Reviewer 3’s Comments:
- The title is not suitable for this manuscript.
We have modified the manuscript accordingly, Line 2.
- I would like the authors to create a hypothesis model using the figure for how the immune response differences from influenza infection between CS and non-CS individuals.
We have adjusted the manuscript to include a summarized hypothesis, Line 525-533.
- In line 176, this is missing “,” between COPD and however.
We have modified the manuscript accordingly, Line 179.
- In line 227, are you sure that this is the H3N3 virus?
We have modified the manuscript accordingly, Line 230-231.
- In line 230, TCID50 should be “TCID50”.
We were not clear on the intentions of the comment. The comment suggests changing to the TCID50 to the same word, Line 233.
- In line 237, “cs exposure” should be “CS exposure”.
We have modified the manuscript accordingly, Line 240.
- In line 256, the reference citation format is not consistent with other references.
We have modified the manuscript accordingly, Line 259.
- Keep the consistent word type throughout the entire manuscript.
We have modified the manuscript to be more consistent.

Round 2
Reviewer 2 Report
Thank you for addressing the comments.